# Family Socio-Economic Status and Children’s Play Behaviors: The Mediating Role of Home Environment

**DOI:** 10.3390/children9091385

**Published:** 2022-09-14

**Authors:** Sixian Li, Jin Sun, Jingxuan Dong

**Affiliations:** 1School of Education, South China Normal University, Guangzhou 510631, China; 2Department of Early Childhood Education, The Education University of Hong Kong, Taipo 999077, Hong Kong

**Keywords:** family socio-economic status, home environment, children’s play behaviors, Chinese

## Abstract

Family socio-economic status (SES) is a significant predictor of children’s early learning performance, while little is known about the relationship between family SES and children’s play. This study aimed to examine how family SES was related to different aspects of children’s play behaviors and whether the home environment served as a mediator in this relationship. A total of 844 mothers of children aged three to six (*N*_boys_ = 431) from different SES backgrounds (*N_low-SES_* = 123, *N_medium-SES_* = 322, *N_high-SES_* = 399) reported the situation of the home environment and their children’s play behaviors with self-developed questionnaires. Results of regression analyses showed that family SES significantly predicted the level of Imagination, Approaches to Learning, and Emotion Expression in children’s play and that the home environment partially mediated such relationships. The results indicate SES-related differences in children’s play behaviors and offer the possibility of narrowing such discrepancies by establishing a child-friendly home environment.

## 1. Introduction

Play is the primary approach to young children’s learning and performs a unique role in early childhood development [1,2,3]. According to Van Oers [4], play is featured as children’s active engagement, allowing children to enjoy a high degree of freedom, and supporting children to follow rules specified by themselves in the activities. Therefore, children’s autonomy and enjoyment are highlighted in the play process. Research has evidenced that play supports children’s cognitive development [5], language performance [6], executive functions [7], and socio-emotional competence [8,9]. More importantly, not only that play contributes to children’s development, but caregivers who participate in play benefit from playing with children. For example, it is found that parents tend to experience a lower level of parental stress and enjoy a more stable and close parent-child relationship than others if they actively play with children, listen to children’s points of view in play, and appreciate children’s humor and independence [10,11]. Therefore, it is meaningful to investigate the influencing factors of children’s play in order to support their play behaviors. This study focused on how children’s play behaviors were related to some key family-related variables, i.e., family socio-economic status (SES) and home environment.

### 1.1. Understanding Children’s Behaviors in Play

Children display a variety of abilities in play, which are important indicators of their development [12]. In other words, the behavior of children during play reflects their social-emotional development, persistence, imagination, and creativity [13] (pp. 220–239). However, the classical measurements of children’s play behaviors, such as the Stages of Play Scale [14], the Penn Interactive Peer Play Scale [15], and the Preschool Play Behavior Scale [16], only focus on children’s social-emotional competencies as reflected in the play process from the aspects of social participation, peer interaction, or solitary behaviors.

Indeed, understanding children’s play behaviors from different aspects can better portray children’s learning and development in multiple domains during the play process, which has been adopted in more recent studies. For instance, The Play Observation Scale (POS) [17] investigates children’s cognitive and social competence as shown during play. The Affect in Play Scale–Preschool Version (APS-P) [18] examines children’s cognitive and emotional development in pretend play by examining their imagination, organization, enjoyment, and emotional expression during the play process.

Chinese society highly emphasizes children’s academic achievement [19] and thus considers play less valuable as said in the Chinese proverb “Learning is meritorious, whereas playing is useless”. In a study with Asian and Western parents on their attitudes towards child play, most Asian parents believe that early learning is much more important than play in facilitating children’s development and they do not believe that play is helpful for children’s school readiness, while the Western parents tend to believe that children learn through play [20]. In the current highly competitive Chinese society, parents focus greatly on children’s academic achievement even in early childhood by sending children to various extra-curricular classes but do not offer children sufficient time for free play [21].

To better support children’s early learning and development through play, Guidelines for Kindergarten Education (trial version) explicitly states that kindergartens in China should adopt play as the primary teaching approach [22]. It is therefore meaningful to better understand Chinese children’s play behaviors from different aspects. This study examined Chinese children’s behaviors in play as reflected in various key developmental domains and further explored how children’s play behaviors were related to two key family-related variables, family SES and home environment.

### 1.2. Family SES and Children’s Play

There has been a long history of studying the relationship between family SES and children’s play. An influential study conducted by Rubin et al. [14] found that compared with children from middle-class families, those from lower-SES families were more likely to be engaged in parallel play but less in associative or cooperative play; these children were also engaged in more functional play, less constructive or sociodramatic play than those from the middle-class families. It was believed that the lower level of pretend play behaviors of children from low-SES families was probably the result of the less responsive parent-children interactions at home and the limited resources available to support children’s learning [23] (pp. 154–196). The SES-related differences in children’s cognitive development as identified in prior studies [24] may also contribute to the complexity levels of play engaged in by children from different backgrounds.

Similar findings were achieved in Mohan and Bhat’s [25] study, in which family SES was found to be significantly associated with the types and proficiency of children’s play. They argued that children from middle-SES families were engaged in more imaginative and cooperative play, compared with their counterparts from low-SES families. Family income is also found to be predictive of the frequency of children’s outdoor play. Compared with children from low-income families, those from higher-income families are found to have more outdoor play activities [26]. This might be related to the location of the family residence, as children from low-income families usually live in communities with fewer parks, more traffic, and higher crime rates, which limit the possibilities for young children to have outdoor play [27].

Despite the evidence showing the relationship between family SES and children’s play, most of the existing studies are focused on how family SES predicts the type of children’s play but little is known about how family SES is related to children’s play behaviors. Therefore, in this study, we investigated how family SES was related to different aspects of children’s play behaviors in the Chinese context and further explored the underlying mechanism by examining the mediating role of the home environment in these relationships.

### 1.3. Home Environment and Children’s Play

According to the Ecological Systems theory [28], children’s psychological and behavioral development is the result of their interactions with the environment. As a microsystem, the family is the place where children first learn socialization and play. The physical surroundings and psychological environment are considered two basic components of the home environment [29], as reflected in the classical Home Observation for Measurement of the Environment (HOME) Inventory [30].

Both the physical and psychological environments at home are predictive of children’s early development. For example, the number of books at home is related to children’s cognitive development [31], and it is found that responsive parent-child interactions support children’s mathematics learning [32]. Cote and Bornstein [33] found that children presented complex and delicate play behaviors if they were supported by mothers’ demonstration and encouragement of autonomy in the play process. Similarly, Fogle and Mendez [34] revealed that parents’ recognition of the value of play was positively related to children’s creative, cooperative, and interactive behaviors during play.

There is plenty of research supporting the relationship between family SES and the home environment. For example, high-SES families usually invest more in children’s learning and development so that children from these families tend to enjoy more learning resources and opportunities than those from low-SES families; there are also more quality parent-child interactions in high-SES families [35,36]. It was also shown that medium- and high-SES parents are more likely to recognize the value of play and communicate more with children than other parents [23].

Due to the relationship identified between the home environment, family SES, and the development of children’s play behaviors, we considered that the home environment may partially explain the relationship between family SES and children’s play behaviors. Santos et al. [37] observed children’s behaviors in social play and found that those from high-SES families were more likely to be raised by authoritative parents so they displayed less destructive but more cooperative behaviors during play and showed a stronger sense of curiosity and creativity. However, there is still a lack of empirical evidence identifying the relationship between family SES and children’s play behaviors in different aspects as well as the potential mediating role of the home environment in these relationships. Consequently, this study aimed to fill in this gap. It was hypothesized that family SES could significantly predict children’s behaviors as reflected in different aspects and the home environment would at least partially mediate the relationship as identified above.

## 2. Materials and Methods

### 2.1. Participants

We randomly selected five public kindergartens in Guangzhou, China, to participate in this study and obtained the consent from these kindergartens. All parents in the five kindergartens were invited and 844 out of 880 parents filled in the consent forms and returned the questionnaires. The number of valid questionnaires collected in each kindergarten ranges from 140 to 204. The details of the participants can be found in Table 1.

### 2.2. Measures

#### 2.2.1. Family SES

We used parental occupation, parental education, and family income reported by the parents to generate the index of family SES. Five categories, i.e., unskilled/semi-skilled workers (1), skilled workers (2), semi-professional/general clerical workers (3), professional/administrator (4), and major professional/higher executive (5) were used to classify parental occupation [38,39]. Parental education was also classified into five categories from “middle school and below (1)” to “master and above (5)”. Family monthly income consisted of six categories from “4000 and below (1)” to “20,000 and above (6)” and was scored from 1 to 6, accordingly. The principal component analysis was adopted to generate the family SES index score. The first component score, which explained 59.3% of the variance, was extracted and used in the following analyses.

#### 2.2.2. Home Environment

Items reflecting the physical and psychological environment at home, particularly the subscales of Parents’ Emotional and Verbal Responsiveness, Maternal Involvement with the Child, Varieties in Daily Stimulation from The HOME Inventory [29], and the items in the subscales of Closeness, Emotional Expression, and Organization from the Family Environment Scale [40] (p. 82) composed the initial questionnaire on the home environment to be used in this study. We also included items related to parents’ beliefs on children’s play as prior studies demonstrated that the parents’ play belief was closely related to children’s play behaviors [34,41]. Therefore, the initial questionnaire on home environment included 35 items. Likert’s 5-point scale from “strongly disagree” to “strongly agree” was adopted and a higher score indicated a better home environment.

Exploratory factor analysis (EFA) identified a six-factor structure. A total of 27 items whose factor loadings were higher than 0.5 were retained to form the Home Environment Questionnaire. Confirmatory factor analysis (CFA) was conducted with another 329 parents and showed a good model fit of the six-factor structure (x^2^/df = 2.230, CFI = 0.916, RMSEA = 0.064, RMR = 0.034, GFI = 0.854, AGFI = 0.821, IFI = 0.917). The analyses in this study, therefore, were based on the 27-item scale (Cronbach’s alpha = 0.918).

The first factor was composed of six items (α = 0.883), which tapped family harmony and emotional expression (e.g., We rarely quarrel nor fight with others in the family.) and was named “Family Atmosphere”. The second factor, which was named “Parenting Style”, included six items (α = 0.896). These items were related to the approaches parents used in parenting practices (e.g., I allow my children to express themselves even if they have different opinions from mine). The third factor “Play Belief” included four items (α = 0.877) (e.g., I believe that play is an important way of learning for kids). The fourth factor was named “Family Rules” and included four items (α = 0.854). It covered items related to the rules followed by family members (e.g., In my family, we serve distinct roles and everyone has his/her own job to do.). The fifth factor was related to parents’ investment and help in children’s learning (e.g., We often teach our kids such as identifying pinyin, Chinese characters, and the number of objects). We named this factor “Learning Support”, which included four items (α = 0.768). The sixth factor “Family Values”, with three items (α = 0.803), measured the value of family and the level of mutual support among family members (e.g., I believe that blood is thicker than water; family is the most important). We summed up each factor score and used the total score as the index of the home environment in the following analyses.

#### 2.2.3. Children’s Play Behaviors

We developed a 21-item Likert’s 5-point scale describing children’s behaviors related to their cognitive, emotional, and social development in play and this formed the Children’s Play Behavior Scale. Parents were required to rate different behaviors in the play of their children from “strongly disagree (1)” to “strongly agree (5)”.

Similarly, EFA was first used to establish the structure of this scale and a four-factor structure was identified. We kept 15 items whose factor loading was higher than 0.5 to form the scale to be used in this analysis (α = 0.933). The four-factor structure was further validated with CFA in another 329 sample of parents (x^2^/df = 2.963, CFI = 0.954, RMSEA = 0.077, RMR = 0.036, GFI = 0.91, AGFI = 0.872, IFI = 0.954).

The first factor “Imagination” included three items (α = 0.867) that reflected children’s ability to perform object substitution and situated imagination, etc. (e.g., The child treats one object as another, such as using blocks as “a phone”, or using a paper box as “a microwave”). The factor of “Approaches to Learning” included three items (α = 0.838) that measured children’s persistence and sustained attention in play (e.g., The child can stay in a game for a long time.). The third factor “Sociality” included five items about children’s cooperation, communication, and problem-solving abilities (α = 0.897) (e.g., When the child conflicts with his/her peers, he/she solves the problem through discussion, negotiation, etc). The fourth factor “Emotion Expression” reflects children’s positive emotion expression and communications in play and included four items (α = 0.914) (e.g., The child claps and dances when he/she is happy and/or excited in play.).

#### 2.2.4. Control Variables

Parents reported basic demographic information, including child age and gender, in the parent questionnaire. We included these variables as control variables in the analyses as both child age and gender were found to be related to children’s play behaviors and home environment in prior studies.

## 3. Statistical Analysis

We first conducted descriptive analyses to understand the distribution and descriptives of key outcome variables (Imagination, Approaches to Learning, Sociality, and Emotion Expression). The domain total scores were used in the analyses. A series of hierarchical regressions was then conducted to understand the family SES-related differences in these four dimensions of play behaviors. The potential mediating role of family environment, indexed as the sum of the six factor scores, was further examined with the bootstrap estimation (bootstrap = 5000) with 95% confidence intervals with AMOS 26.0 in the significant relationships identified between Family SES and children’s play behaviors.

## 4. Results

### 4.1. Descriptives

Table 2 presents the descriptive statistics of children’s play behaviors, family SES, and home environment. We further explored whether there were significant age and gender differences in children’s Imagination, Approaches to Learning, Sociality, and Emotion Expression in play. Results of MANOVA showed significant differences related to child age in all four dimensions of play behaviors (Imagination: *F* = 5.572, *p* = 0.001, partial *η^2^* = 0.020; Approaches to Learning: *F* = 5.562, *p* = 0.001, partial *η^2^* = 0.020); Sociality: *F* = 11.408, *p* = 0.000, partial *η^2^* = 0.039; Emotion Expression: *F* = 3.846, *p* = 0.009, partial *η^2^* = 0.014). Follow-up post-hoc comparisons showed that three-year-olds’ performances were significantly lower in all four dimensions than older children (*p* < 0.05), while no significant age-related differences were revealed among other age groups. No significant gender differences or Age × Gender differences were detected.

### 4.2. Family SES, Home Environment, and Children’s Play Behaviors

As shown in Table 2, child age, family SES, and home environment, were all significantly correlated with children’s Imagination, Approaches to Learning, Sociality, and Emotion Expression during play. Child gender was only significantly correlated with Approaches to Learning during play.

Controlling for child age and gender, hierarchical regression analyses further revealed that family SES significantly predicted Imagination (*β* = 0.128, *p* < 0.001), Approaches to Learning (*β* = 0.119, *p* < 0.001), and Emotion Expression (*β* = 0.067, *p* < 0.05), while home environment significantly predicted children’s Imagination (*β* = 0.372, *p* < 0.001), Approaches to Learning (*β* = 0.451, *p* < 0.01), Sociality (*β* = 0.426, *p* < 0.001), and Emotion Expression (*β* = 0.375, *p* < 0.001) (see Table 3).

We further examined the potential mediating effects of the home environment in the relationship between family SES and children’s Imagination, Approaches to Learning, and Emotion Expression in play, respectively, with AMOS 26.0. The sensitivity analysis was conducted using the bootstrap estimation (bootstrap = 5000) with 95% confidence intervals. The results indicated that the home environment was a significant mediator, which partially mediated the relationship between family SES and children’s Imagination, Approaches to Learning, and Emotion Expression. The path coefficients and model fit index can be found in Table 4.

## 5. Discussion

This study examined the relationship between family SES and children’s play behaviors from four different aspects, namely, Imagination, Approaches to Learning, Sociality, and Emotion Expression in a city in China. The mediating role of the home environment in these relationships was further investigated.

### 5.1. Characteristics of Children’s Play Behaviors

We found significant age differences in children’s Imagination, Approaches to Learning, Sociality, and Emotion Expression in play, and these differences were only discovered between the three-year-olds and the older groups.

This, on the one hand, might be related to the first transitional environment: from home to the kindergarten. For young children at age three, it is a challenging task for them to adapt to a new environment [42] (pp. 942–955). After one year of adaptation, children get used to the kindergarten setting and benefit from the rich learning resources, activities, and social environment provided. It is therefore not surprising that starting from age four, children exhibit significantly better performance in all aspects of play behaviors than children of age three.

On the other hand, the age-related differences in play behaviors, especially the differences between three-year-olds and those above, are closely related to the developmental stages of children’s overall competence. For example, the ability to make representation lays the foundation for imagination in play, and this competence is usually found to occur at around age four [43] (pp. 141–160) [44]. In addition, the period of 33 to 45 months is a crucial period for children to develop approaches to learning [45]. This is consistent with the finding related to children’s approaches to learning shown in play by children aged three and above. The rapid growth of language ability and theory of mind around age four [46,47] might lead to more positive and appropriate expression of emotion and better social competence for children aged four or above compared with the three-year-olds.

Nevertheless, it was found there were no significant differences in the four aspects of children’s play behaviors among children aged four and above. This might be related to the fact that Chinese kindergartens provide similar play support for children of all ages. For example, the play materials for children’s free play are generally the same in all grades and the setup of classroom learning corners is largely invariant [48]. These may limit the diversity and fruitfulness of children’s play behaviors. Although early years are critical for children’s development in either cognitive, language, or social-emotional domains [49], children seem not to obtain sufficient support from their free play activities in the kindergarten setting so they were not capable to employ their competence in play. Meanwhile, due to the great importance attached to children’s academic performance in Chinese culture, parents, especially those in urban areas, usually have their children attend extracurricular classes as the children are older [50,51,52]. Such arrangements decrease children’s limited playtime, reducing the opportunities for them to play to the full extent, which could also explain the lack of age differences found in play behaviors among children aged four and older.

We did not find any significant gender differences in children’s play behaviors, which might be explained by the undifferentiated education in kindergarten, as well as the equivalent parenting practices for boys and girls in urban China. Some studies suggest that boys and girls have different preferences in the types of play. For example, boys prefer car toys and are fond of sports and adventure games, while girls prefer dolls for role play [53,54]. Nevertheless, it is unlikely that the gender-related preferences in the type of play lead to differences in children’s play behaviors as reflected in the aspects of Imagination, Approaches to Learning, Sociality, and Emotion Expression. Further empirical studies are needed to support such inference.

### 5.2. Family-SES-Related Differences in Children’s Play Behaviors

Based on the survey of 844 urban parents, this study found that family SES was significantly associated with children’s Imagination, Approaches to Learning, and Emotion Expression, but not Sociality, in play.

Such findings are in line with our hypothesis to some extent. Parents from high-SES families not only invest more time and resources in supporting children’s early learning and development [55,56] but also have higher quality interactions with them [57], which are beneficial for children’s cognitive and socio-emotional development [58,59]. Because of the more sufficient educational investment and quality parent-child interactions, children from high-SES families can use play materials more freely and better enjoy play, which supports their imagination, approaches to learning, and appropriate emotion expression. In fact, previous studies have similar findings. For instance, Harvey et al. [60] found that high-SES children were less restricted by their parents in free play and showed stronger imagination in play.

Although a large number of studies have pointed out that children from low-SES families have lower social skills than those from high-SES families [59,61], we did not find significant family-SES differences in children’s play behaviors related to Sociality, as reported by their mothers. On the one hand, it reminds us to reflect on whether children’s social skills in play reflect their actual competence, as the level of social skills children exhibited in play is largely related to the play materials offered and the social opportunities available. On the other hand, this might be related to similar opportunities for them to socialize both at home and in kindergarten. Although the “one-child policy” was rescinded in 2015, families with more than one child are still not common in urban China as is reflected in the sample of this study. The majority of the families in our sample only had one child and the peer group at kindergarten was the primary socialization agent for these children [62] (pp. 11–29). Despite the differences in educational quality revealed in kindergartens, the socialization opportunities for children there do not differ significantly [63]. Similar socialization opportunities both at home and in kindergarten for children from different family backgrounds might help explain the similar level of social competence displayed in play.

### 5.3. Home Environment and Children’s Play Behaviors

It is found that the home environment, as an integration of physical surroundings and psychological environment, significantly predicted children’s Imagination, Approaches to Learning, Emotion Expression, and Sociality in play, which conforms to the findings of previous studies on the value of quality home environment in the early development of children [64].

In this study, such family values and emotion elements as parents’ beliefs on child play, family atmosphere, and family values that were not commonly considered in prior studies were taken into account. The positive relationship found between the home environment and children’s play behaviors suggests that besides the physical resources and positive parent-child relationship, a harmonious family atmosphere and recognition of the value of early years are also important for early childhood development, especially for children’s play behaviors. When families are warm and supportive, children are found to be more self-determinant and confident [65], which is important for children to demonstrate their imagination, persistence, positive emotion expression, and social competence during their play.

As expected, the home environment was found to mediate the relationship between family SES and children’s Imagination, Approaches to Learning, and Emotion Expression in play. In other words, such factors as the education investment [66], quality parent-child interactions [67,68,69,70,71], the recognition of family values and rules [56,72], and families’ atmosphere, which are highly related to family SES, partially explain the mechanism of how family SES influence children’s play behaviors. If we want to narrow the SES-related gaps in children’s play behaviors and to further support children’s learning through play, more attention should be paid to the establishment of a quality home environment, especially in low-SES families.

## 6. Limitations and Implications

This study has the following limitations. First of all, all analyses were based on parents’ self-reported data, which were likely to incur bias from the reporters [73,74,75]. Although parents were explicitly told that their answers had nothing to do with right or wrong and would not lead to any consequences for the children, we could not avoid the tendency of social pleasing from parents and were not able to ensure parents’ understanding of the questions asked in the questionnaire. Therefore, data collected from multiple sources, such as observation of children’s play behavior, will be valuable in future studies. Second, the five public kindergartens sampled in this study were all of relatively high quality and in a rather developed city in China. Although the parents reported great differences related to family SES, they were not representative of the diverse families in the country. Future studies can recruit samples from regions with more diverse levels of economic development and reflect greater differences in family SES. Third, this study did not examine the individual (e.g., children’s cognitive and social competencies), family (e.g., children’s play experiences at home), and school factors (e.g., play support from the kindergarten) that were closely related to children’s play behaviors. Therefore, it is necessary for follow-up studies to include these factors, so as to obtain a more comprehensive understanding of the relationship between family SES and children’s play behaviors.

Nevertheless, this study has profound theoretical and practical implications. First of all, the study investigated children’s play behaviors from the aspects of Imagination, Approaches to Learning, Emotion Expression, and Sociality, which has furthered the understanding of how play supports children’s overall development. Besides the learning support provided at home, the study also examined other factors related to children’s home environment, such as family values and family atmosphere, and found its predictive effects on children’s play behaviors. This will be implicative for a better conceptualization of the home environment for young children. The findings are also of practical implications. For example, no significant age-related differences were found in the play behaviors among children aged 4 to 6, which reminds educators and parents to reflect on the sufficiency and appropriateness of the support provided for these children in relation to their play at home, and in kindergarten settings. Finally, the identification of the relationships between family SES and children’s play behaviors in Imagination, Approaches to Learning, and Emotion Expression, and the mediating role of the home environment in these relationships suggests more support in the children’s home environment is needed to support low-SES children in their early play and learning.

## Figures and Tables

**Table 1 children-09-01385-t001:** Demographic Characteristics of the Sample.

		Age 3 (n = 199)		Age 4 (n = 314)		Age 5 (n = 255)		Age 6 (n = 76)	
		n	%	n	%	n	%	n	%
Gender	Male	99	49.7	162	51.6	133	52.2	37	48.7
	Female	100	50.3	152	48.4	122	47.8	39	51.3
Father’s age	25 and under	2	1	0	0	1	0.4	0	0
	26–30	15	7.5	19	6.1	8	3.1	2	2.6
	31–40	130	65.3	218	69.4	179	70.2	54	71.1
	41–50	52	26.1	73	23.2	64	25.1	19	25
	51 and above	0	0	4	1.3	3	1.2	1	1.3
Mother’s age	25 and under	2	1	2	0.6	1	0.4	0	0
	26–30	30	15.1	37	11.8	17	6.7	10	13.2
	31–40	137	68.8	229	72.9	209	82	56	73.7
	41–50	30	15.1	45	14.3	28	11	10	13.2
	51 and above	0	0	1	0.3	0	0	0	0
Paternal education	Middle school and below	9	4.5	5	1.6	9	3.5	3	3.9
	High school/Secondary/technical school	32	16.1	32	10.2	27	10.6	4	5.3
	Post-secondary technical school	32	16.1	53	16.9	48	18.8	20	26.3
	Bachelor	95	47.7	164	52.2	137	53.7	38	50
	Master and above	31	15.6	60	19.1	34	13.3	11	14.5
Maternal education	Middle school and below	8	4	5	1.6	7	2.7	3	3.9
	High school/Secondary/technical school	27	13.6	38	12.1	14	5.5	9	11.8
	Post-secondary technical school	39	19.6	63	20.1	64	25.1	22	28.9
	Bachelor	100	50.3	159	50.6	141	55.3	36	47.4
	Master and above	25	12.6	49	15.6	29	11.4	6	7.9
Paternal occupation	Unskilled/semi-skilled worker	23	11.6	24	7.6	35	13.7	8	10.5
	Skilled worker	28	14.1	48	15.3	42	16.5	14	18.4
	Semi-professional/general clerical worker	44	22.1	50	15.9	44	17.3	15	19.7
	Professional/administrator	75	37.7	130	41.4	99	38.8	30	39.5
	Major professional/higher executive	29	14.6	62	19.7	35	13.7	9	11.8
Maternal education	Unskilled/semi-skilled worker	36	18.1	56	17.8	52	20.4	18	23.7
	Skilled worker	28	14.1	24	7.6	32	12.5	8	10.5
	Semi-professional/general clerical worker	41	20.6	82	26.1	66	25.9	15	19.7
	Professional/administrator	79	39.7	117	37.3	82	32.2	29	38.2
	Major professional/higher executive	15	7.5	35	11.1	23	9	6	7.9
Family monthly income	¥4000 and below	6	3	3	1	6	2.4	1	1.3
	¥4001–¥7000	22	11.1	16	5.1	14	5.5	7	9.2
	¥7001–¥10,000	32	16.1	39	12.4	33	12.9	9	11.8
	¥10,001–¥15,000	37	18.6	74	23.6	67	26.3	14	18.4
	¥15,001–¥20,000	46	23.1	69	22	56	22	13	17.1
	¥20,000 and above	56	28.1	113	36	79	31	32	42.1

**Table 2 children-09-01385-t002:** Descriptive Statistics and Correlations of Study Variables ^1^.

	Gender	Age	Family SES	Home Environment	Imagination	Approaches to Learning	Sociality	Emotion Expression
Gender	1							
Age	−0.01	1						
Family SES	−0.07 *	−0.02	1					
Home environment	−0.09 **	0.07 *	0.25 **	1				
Imagination	0.03	0.12 **	0.22 **	0.40 **	1			
Approaches to Learning	−0.09 **	0.11 **	0.23 **	0.49 **	0.65 **	1		
Sociality	−0.00	0.18 **	0.12 **	0.44 **	0.43 **	0.60 **	1	
Emotion Expression	−0.05	0.08 *	0.16 **	0.40 **	0.46 **	0.54 **	0.50 **	1
Mean	1.49	4.25	18.319	4.267	4.012	3.960	3.756	4.269
SD	0.500	0.913	4.175	0.480	0.756	0.681	0.654	0.626

^1^ * *p* < 0.05, ** *p* < 0.01.

**Table 3 children-09-01385-t003:** Regression Coefficients of Family SES and Home Environment on Children’s Play Behaviors ^1^.

Variable	Imagination			Approaches to Learning			Sociality			Emotion Expression		
	B	β	SE	B	β	SE	B	β	SE	B	β	SE
Constant	0.63		0.25	0.76		0.22	0.72		0.21	1.88		0.21
Gender	0.12	0.08 *	0.05	−0.05	−0.04	0.04	0.05	0.04	0.04	−0.01	−0.01	0.04
Age	0.08	0.10 **	0.03	0.06	0.08 *	0.02	0.11	0.15 ***	0.02	0.04	0.05	0.02
SES	0.02	0.13 ***	0.01	0.02	0.12 **	0.00	0.02	0.01	0.00	0.01	0.07 *	0.00
Home Environment	0.59	0.37 ***	0.05	0.64	0.45 **	0.04	0.58	0.43 ***	0.43	0.49	0.38 ***	0.04
R2			0.19			0.26			0.21			0.17
△R2			0.19			0.26			0.21			0.16

^1^ * *p* < 0.05, ** *p* < 0.01, *** *p* < 0.001.

**Table 4 children-09-01385-t004:** Mediating Models Linking Family SES, Home Environment, and Children’s Play Behaviors.

Play Behavior	Path	Coefficient	SE	95% CI		Variance Explained
				LL	UL	
Imagination	Family SES → Imagination	0.12 ***	0.03	0.06	0.19	50%
	Family SES → Home environment	0.25 ***	0.04	0.18	0.33	
	Home environment → Imagination	0.37 ***	0.04	0.29	0.45	
	Family SES → Home Environment → Imagination	0.02 *	0.00	0.01	0.02	50%
Model Fit Index	x^2^/df = 6.153, RMSEA = 0.078, IFI = 0.979, CFI = 0.979, GFI = 0.997, AGFI = 0.956					
Approaches to Learning	Family SES → Approaches to Learning	0.12 **	0.03	0.01	0.13	58.8%
	Family SES → Home Environment	0.25 ***	0.04	0.18	0.33	
	Home environment → Approaches to Learning	0.45 ***	0.03	0.39	0.51	
	Family SES → Home environment → Approaches to Learning	0.02 *	0.03	0.01	0.02	41.2%
Model Fit Index	x^2^/df = 1.740, RMSEA = 0.030, IFI = 0.998, CFI = 0.998, GFI = 0.999, AGFI = 0.988					
Emotional Expression	Family SES → Emotional Expression	0.07 *	0.04	0.00	0.14	41.9%
	Family SES → Home environment	0.25 **	0.04	0.18	0.33	
	Home environment → Emotional Expression	0.38 ***	0.04	0.31	0.44	
	Family SES → Home environment → Emotional Expression	0.01 *	0.00	0.01	0.02	58.1%
Model Fit Index	x^2^/df = 1.521, RMSEA = 0.025, IFI = 0.995, CFI = 0.995, GFI = 0.999, AGFI = 0.989					

* *p* < 0.05, ** *p* < 0.01, *** *p* < 0.001.

## Data Availability

Data available on request due to restrictions eg privacy or ethical.

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
