# Peer review of "Family Socio-Economic Status and Children’s Play Behaviors: The Mediating Role of Home Environment"

_children, 2022, doi:10.3390/children9091385_

Round 1

Reviewer 1 Report

Dear Editor,

Thank you for the opportunity to review the manuscript. The study is very interesting. However, it is heavy to read. Overall there is too much information and it is not clear whether the objective of the study has been achieved. The authors should summarise a little by removing unnecessary information.

- Why Family socioeconomic status is abbreviated using SES? If you want to use this abbreviation you should separate socio-economic with a hyphen

-        Line 21: "Keywords: family socioeconomic status; home environment; children's play behaviours". These are all keywords already present in the title. You should find other keywords to give your manuscript a better chance of being searched.

-        Line 43: (pp.220-239) can be omitted. Correct in the rest of manuscript.

-        The statistical analysis section is missing in the methods. It is not clear what was done with the questionnaires. You should summarise these informations in this section, so that all the statistical tools used are explained (MANOVA, regression ecc..).

-     Review the style of the references as there are small errors. Example before the year once you use a full stop and once a comma. 

    Conclusions here would be advisable as there is too much information and it would be useful to summarise the results with short sentences.

     Good work!

Author Response

Thank you for the very detailed comments and suggestions. We have addressed all the suggestions in the revised manuscript. 

  1. We have separated socio-economic with a hyphen as suggested throughout the manuscript;
  2. We have updated the keywords list as suggested to reflect the contents of this manuscript better;
  3. We have added a statistical analysis section as suggested;
  4. We have reviewed and checked the reference list and format to correct the typos; 
  5. We have tried to streamline and better summarize the results and conclusions of this study with short sentences. 

Reviewer 2 Report

Dear authors,

The article discusses an important topic - the relationship between family and child and how this influences the psycho-intellectual development of children.

The article focuses on the description of the situation in Chinese society, but it would be very interesting if you would pay attention to the characteristics of Chinese society compared to other types of societies/countries. I believe that this aspect would strengthen the manuscript's originality.

Best regards,

Author Response

Thanks for the insightful suggestion. We have included a paragraph on page 2 about Chinese parents' perception and understanding of child play compared to parents in other countries.  

Round 2

Reviewer 1 Report

The statistical analysis section is part of the method, so you should title it 2.3 Statistical Analysis. Otherwise, the changes have been made.